# Floral Scents in Bee-Pollinated Buckwheat and Oilseed Rape under a Global Warming Scenario

**DOI:** 10.3390/insects14030242

**Published:** 2023-02-28

**Authors:** Guaraci Duran Cordeiro, Stefan Dötterl

**Affiliations:** Department of Environment & Biodiversity, Paris-Lodron University of Salzburg, Hellbrunnerstr. 34, 5020 Salzburg, Austria

**Keywords:** chemical communication, climate change, pollination, buckwheat, oilseed rape, honeybee, bumblebee

## Abstract

**Simple Summary:**

Global warming is expected to impact the communication between flowering plants and their pollinators. We aimed to test how increased temperatures affect the chemical signaling between two important crop species (buckwheat, *Fagopyrum esculentum*; oilseed rape, *Brassica napus*) and their bee pollinators (*Apis mellifera, Bombus terrestris*). Floral scent analyses showed that buckwheat was affected by increased temperatures, whereas in oilseed rape, both total scent emission and scent composition were independent of temperature. The floral scent of oilseed rape was dominated by *p*-anisaldehyde and linalool at both temperatures tested. Buckwheat emitted threefold less floral scent and a different composition at warmer temperatures. Some compounds, among them linalool and indole, were only released from buckwheat plants cultivated at optimum temperatures but not from plants cultivated at warmer temperatures; however, 2- and 3-methylbutanoic acid were the most abundant compounds at both temperature regimes. The bees detected many floral scent compounds of buckwheat in electroantennographic analyses, among them compounds that disappeared at warmer temperatures. Our study highlights that oilseed rape is more heat tolerant and resilient than buckwheat and that the temperature-induced scent changes in buckwheat affect the olfactory perception of the flowers by bees.

**Abstract:**

Many wild plants and crops are pollinated by insects, which often use floral scents to locate their host plants. The production and emission of floral scents are temperature-dependent; however, little is known about how global warming affects scent emissions and the attraction of pollinators. We used a combination of chemical analytical and electrophysiological approaches to quantify the influence of a global warming scenario (+5 °C in this century) on the floral scent emissions of two important crop species, i.e., buckwheat (*Fagopyrum esculentum*) and oilseed rape (*Brassica napus*), and to test whether compounds that are potentially different between the treatments can be detected by their bee pollinators (*Apis mellifera* and *Bombus terrestris*). We found that only buckwheat was affected by increased temperatures. Independent of temperature, the scent of oilseed rape was dominated by *p*-anisaldehyde and linalool, with no differences in relative scent composition and the total amount of scent. Buckwheat emitted 2.4 ng of scent per flower and hour at optimal temperatures, dominated by 2- and 3-methylbutanoic acid (46%) and linalool (10%), and at warmer temperatures threefold less scent (0.7 ng/flower/hour), with increased contributions of 2- and 3-methylbutanoic acid (73%) to the total scent and linalool and other compounds being absent. The antennae of the pollinators responded to various buckwheat floral scent compounds, among them compounds that disappeared at increased temperatures or were affected in their (relative) amounts. Our results highlight that increased temperatures differentially affect floral scent emissions of crop plants and that, in buckwheat, the temperature-induced changes in floral scent emissions affect the olfactory perception of the flowers by bees. Future studies should test whether these differences in olfactory perception translate into different attractiveness of buckwheat flowers to bees.

## 1. Introduction

Human population growth will demand in this century an increase of c. 70% in food production [1,2,3]. Insects, the most diverse group of organisms in terrestrial ecosystems [4], may be very helpful in reaching this goal, as they are important pollinators. Pollination by insects is a key process in the production of food since they improve the yield and quality of global crop species that are important for the human diet [5,6,7]. Among insects, bees are the most efficient crop pollinators [8,9]. Although pollination by bees is an essential ecosystem service, little is known about the attraction of bees to flowers of crop plants and the impact of global warming on these interactions [10,11,12,13,14].

The communication between flowers and bees precedes pollination, making floral signaling an essential step in the production of food. Among floral traits involved in communication with pollinators, scents play a key role in the attraction of bee pollinators [10,13,15], providing information about the identity of the plant species and the location, abundance and quality of floral rewards [16,17,18]. Scents are made up of complex bouquets of volatile organic compounds (VOCs), which belong to several chemical classes [19].

Many different chemicals have been identified from crops to date [12,13,20,21,22,23], and most of them are generally widespread among flowering plants (e.g., benzaldehyde, (*E*)-*β*-ocimene, linalool), whereas some others are very rare among plants (e.g., (*E*)-*N*-(2-methylbutyl)- and (*E*)-*N*-(3-methylbutyl)-1-(pyridin-3-yl)methanimine) [24]. So far, there is a large gap in our understanding of how floral scents of crop plants are affected by climate warming. Studies of non-crop species, however, have shown that qualitative and quantitative floral scent properties are affected by increased temperatures [25,26,27,28,29], and a very recent study on strawberry even demonstrated that flowers stop producing detectable amounts of scent when grown at temperatures 5 °C higher than optimal temperatures ([30]). These changes in floral scent may cause inefficient pollinator attraction [31,32,33,34] in the future, considering that global mean surface temperatures are increasing [35]. However, little is known about how temperature-induced changes in scent emission affect pollinator behavior and thus pollination in natural and agricultural settings. Therefore, it is difficult to predict the pollinator-mediated changes in fruit sets caused by climate change.

Here, we evaluated how global warming affects the chemical signaling between two important crop species and their bee pollinators. The target crops were buckwheat (*Fagopyrum esculentum* Moench—Polygonaceae) and oilseed rape (*Brassica napus* L.—Brassicaceae, variety—Sommer-Rape). Buckwheat is a pseudo-cereal that is processed into products such as breakfast foods, flour and noodles. Its pollination is highly dependent on insects, mainly bees [8,36,37]. Oilseed rape is used for the production of edible and fuel oils, with bee pollination significantly promoting seed set and oil content [8,9,38,39], but see [40]. Specifically, we used a combination of chemical analytical and electrophysiological approaches to quantify the influence of the worst-case global warming scenario (+5 °C in this century; SSP-8.5) [35] on floral scent emissions and on the olfactory detection of these plants by their main managed bee pollinators (*Apis mellifera* Linnaeus and *Bombus terrestris* Linnaeus).

## 2. Material and Methods

### 2.1. Crop Plant Cultivation and Temperature Regime

The plants were cultivated in two plant growth chambers (Liebherr, Profi line, Germany; adapted with a multistage temperature controller, model TAR 1700-2, Elreha, Germany; and a light timer switch, model D21ASTRO 230 V 50/60 Hz, Legrand, Germany) that differed in their temperature settings (optimum and increased, see below). The seeds were randomly assigned to one of the chambers, sown in pots (9 × 8 × 9 cm) using standard soil (Einheitserde^®^, Profi Substrat), and fertilized once with 50 mL/pot of fertilizer (Wufax^®^, nitrogen: 12%; phosphate: 4%; potassium: 6%) when the plants reached mid-age [41].

The requirements of the crops for light conditions and water availability were controlled following the data in the literature [42,43]. Plants were cultivated in both chambers at 14-h light/10-h darkness and the light intensity was 2000 lx (via cool white led lamps, model VT-5959 LED-Flutlicht, V-TAC, 50 W). The air humidity ranged between 60% and 70%. To keep the soil at comparable moisture levels during the development of the plants and between the different treatments, as measured by a tensiometer (model FDA 602 TM2, ALMEMO^®^, Germany), the water supply varied according to the age of the plants and the temperature scenario. When the plants were sown, the amount was, independent of the scenario, 15 mL/plant/day; in the mid-age, 60 mL/plant/day (optimum scenario) and 90 mL/plant/day (warmer scenario); during the flowering phase, 120 mL/plant/day (optimum scenario) and 170 mL/plant/day (warmer scenario).

The plants were grown under two temperature scenarios: optimum and 5 °C higher than optimal temperatures (according to the global warming scenario SSP-8.5 [35]). The mean optimal temperature for the growth and flowering of buckwheat is 22 °C [44], and for oilseed rape, it is 20 °C [45]. Considering the mean daily thermal amplitude in Central Europe [46,47], buckwheat plants were cultivated in the optimal scenario at day and night with temperatures of 26 °C and 16 °C (mean 22 °C, considering the length of the day and night periods), respectively, and in the warmer scenario, the temperatures were 31 °C and 21 °C (mean 27 °C) during day and night, respectively. Oilseed rape plants were cultivated in the optimal scenario at 23 °C and 13 °C (mean 20 °C), respectively, and in the warmer scenario at 28 °C and 18 °C (mean 25 °C), respectively. For each temperature scenario, 12 individual plants were cultivated.

### 2.2. Sampling and Analysis of Flower Scents

Sampling of scents was performed inside the growth chambers by dynamic headspace [48]. From 12 individuals cultivated for each crop and scenario, 11 and 7 buckwheat individuals were sampled in the optimum and warmer scenarios, respectively. From oilseed rape, we sampled seven and six individuals in the optimum and warmer scenarios, respectively. The samples were obtained from inflorescences at the beginning of their first day of anthesis. A single inflorescence of buckwheat (N = 6–24 flowers from optimum scenario; N = 6–25 flowers from warmer scenario) and oilseed rape (N = 3–18 flowers from optimum scenario; N = 4–17 flowers from warmer scenario) per sample were enclosed in a polyester oven bag (Toppits^®^). After bagging, two small adsorbent tubes were inserted into the bag: one was used to trap the floral scent and the other (glass vial filled with 5 mg of Carbotrap B) was used to insert clean air from outside the growth chambers to avoid internal air contamination. The samplings using membrane pumps (G12/01 EB; Gardner Denver Thomas GmbH, Fürstenfeldbruck, Germany) lasted 2 h for buckwheat and 1 h for oilseed rape. This time period was enough to obtain the maximum number of compounds as determined by preliminary analyses that used sampling times between 15 min and 2:30 h. The flows of both pumps were adjusted at 200 mL/min with the help of flowmeters. The adsorbent tubes (quartz vials; length: 25 mm; inner diameter: 2 mm) were filled with 1.5 mg of Tenax-TA (mesh 60–80) and 1.5 mg of Carbotrap B (mesh 20–40, both Supelco). The adsorbents were fixed in the tubes using glass wool. Dynamic headspace samples of green leaves (N = 3 samples per scenario and plant species) were collected with the same method as described above to discriminate between vegetative (not considered for subsequent analyses) and flower-specific scent components. 

Scent samples were analyzed using GC/MS (gas chromatography/mass spectrometry) as described previously [49]. The system consisted of an automated thermal desorption system (model TD-20; Shimadzu, Japan) coupled to a QP2010 Ultra EI GC/MS (Shimadzu, Japan) equipped with a Zebron™ ZB-5 fused silica column (5% phenyl, 95% dimethylpolysiloxane; 60 m long; inner diameter 0.25 mm; film thickness 0.25 μm; Phenomenex). The GC/MS data were processed using GCMS solution (version 4.41, Shimadzu Corporation 2015). The tentative identification of compounds was performed using the mass spectral libraries Wiley 9, Nist 2011/FFNSC 2 and [50], as well as the database available in MassFinder 3. If possible, the identity of the compounds was confirmed by comparison of mass spectra and retention times with those of authentic standard compounds available at the Plant Ecology lab of the Paris-Lodron University of Salzburg. To determine the amount of scent trapped, known amounts of monoterpenes, aliphatics and aromatics were added to clean adsorbent tubes and analyzed by GC/MS as described above; the mean peak areas (total ion current) of these compounds were used to determine the total amounts of crop floral scents [22].

### 2.3. Electroantennographic Detection

As the floral scents of buckwheat differed between temperature treatments (see Results), we tested whether these changes affected the perception of the scents by the bees. We used bee pollinators of buckwheat [8] for our experiments: *Apis mellifera* (obtained from a local beekeeper) and *Bombus terrestris* (Biobest^®^, Belgium). We prepared one synthetic mixture that contained most of the compounds released from either temperature scenario and tested it on the antennae of bee pollinators (N = 5 worker bee individuals of *A. mellifera*; N = 12 worker bee individuals of *B. terrestris*) to identify physiologically active compounds following [51]. The synthetic mixture contained 12 compounds: 3-methylbutanoic acid, 2-methylbutanoic acid, indole, linalool, *p*-cresol, *p*-anisaldehyde, sabinene, butanoic acid, *p*-benzoquinone, linalool oxide furanoid (mixture of isomers), *β*-ocimene (mixture of isomers) and terpinene-4-ol. The compounds available in the synthetic mixture explained on average 94% and 100% of the total floral scents collected from buckwheat grown in the optimum and warmer scenarios, respectively. 

The GC/EAD (gas chromatography/electroantennographic detection) system, the same as that used in [52], consisted of a gas chromatograph (Agilent 7890A, Santa Clara, CA, USA) equipped with a flame ionization detector (FID) and an EAD setup (heated transfer line, 2-channel USB acquisition controller) provided by Syntech (Kirchzarten, Germany). A volume of 1 μL of the sample was injected (temperature of injector: 250 °C) splitless at an oven temperature of 40 °C, followed by opening of the split vent after 0.5 min and heating the oven at a rate of 10 °C min^−1^ to 220 °C. A DMT Beta SE column (30 m long; inner diameter 0.25 mm; film thickness 0.23 μm; MEGA-DEX; BGB Analytik Vertrieb GmbH) was used for the analyses and the column flow (carrier gas: hydrogen) was set at 3 mL min^−1^. The column was split at the end by a μFlow splitter (Gerstel, Mülheim, Germany) into 2 deactivated capillaries leading to the FID (2 m × 0.15 μm) and EAD (1 m × 0.2 μm) setups. Makeup gas (N_2_) was introduced into the splitter at 25 mL min^−1^. The outlet of the EAD was placed in a cleaned and humidified airflow that was directed over the antennae of the pollinators. An antenna was cut at its base and tip, inserted between two electrodes filled with an insect ringer (8.0 g/L NaCl, 0.4 g/L KCl, 0.4 g/L CaCl_2_), and connected to silver wires, as described previously [48]. A floral compound was considered EAD-active in a bee species when it elicited a depolarization response in at least four individuals.

### 2.4. Data Analyses

The Mann–Whitney U test (PAST Version 2.17c [53]) and PERMANOVA (based on Bray–Curtis similarities of the percentage contribution of single compounds to total scent, [54]; Primer 6, version 6.1.15 and Permanova version 1.0.5) were used to test for differences in the total amount of scent per flower and relative scent composition, respectively, between the different temperature scenarios. SIMPER was used to determine the compounds most responsible for relative differences in scent between the two temperature scenarios (Primer 6, version 6.1.15). Nonmetric multidimensional scaling (NMDS), again in Primer, based on the Bray-Curtis similarities, was used to graphically display similarities and differences in relative scent compositions among scent samples and between temperature scenarios. 

## 3. Results and Discussion

The scent samples collected from buckwheat inflorescences contained 23 VOCs overall (23 and 11 in the optimum and warmer scenarios, respectively) from 6 chemical classes: monoterpenes (9 compounds), aromatics (5), aliphatics (3), nitrogen-containing compounds (3), C5-branched chain compounds (2) and irregular-terpenes (1) (Table 1). Most of these compounds were not known to be released from the flowers of buckwheat, and only butanoic acid, 2- and 3-methylbutanoic acid, pentanoic acid, *p*-benzoquinone and *α*-farnesene were also identified in a previous study of buckwheat [55]. Among the compounds newly identified for buckwheat are compounds widespread among floral scents (e.g., linalool, (*E*)-*β*-ocimene, phenylacetonitrile, indole) but also rare floral scents (e.g., 3-pyridinecarboxaldehyde, 2-methylpyrazine) [19]. 

Samples collected at warmer temperatures emitted threefold less scent than samples collected in the optimum scenario (0.71 ± 0.28 ng/flower/hour versus 2.38 ± 0.39 ng/flower/hour; Z = −2.89, N = 18, *p* = 0.004; Table 1, Figure 1A). Temperature also affected the scent composition (Pseudo-F_1,17_ = 7.19, *p* = 0.0001), with samples of plants grown at optimal temperatures being dominated by 2- and 3-methylbutanoic acid (together 46%) and linalool (10%). Plants grown at warmer temperatures were even more dominated by 2- and 3-methylbutanoic acid (73%); however, linalool was absent (Table 1, Figure 1B). These three compounds were most responsible for differences in relative scent composition between the two temperature scenarios (Figure 1B).

Among 12 buckwheat compounds tested on the antennae of *A. mellifera* and *B. terrestris*, 11 elicited antennal responses in both species, among them monoterpenes (4 compounds), aliphatics (2), aromatics (2), C5-branched chain compounds (2) and nitrogen-containing compounds (1) (Table 1; Figure 2). Only the monoterpene sabinene did not elicit a physiological response. Thus, bees have the capability to detect compounds that disappear at increased temperatures or are affected in their (relative) amounts. Some of the EAD-active compounds were already identified as being electrophysiologically active in both *A. mellifera* and *B. terrestris* (e.g., linalool, (*E*)-*β*-ocimene, *p*-anisaldehyde [24,30,56,57,58,59]). The main compounds 2- and 3-methylbutanoic acid were demonstrated here for the first time as eliciting antennal responses in bees. These compounds and butanoic acid also elicited an antennal response in an egg parasitoid that visits buckwheat flowers for nectaring, as evidenced in a previous study [55]. Therefore, the temperature-induced changes in the floral scent emission of buckwheat affect the olfactory perception of the flowers by bee pollinators and other flower visitors.

The samples collected from oilseed rape flowers emitted a total of nine VOCs (five and nine in the optimum and warmer scenarios, respectively) of three chemical classes: monoterpenes (six compounds), aromatics (two) and aliphatics (one) (Table 1). We found a smaller number of compounds in this species than in other studies, and most of the compounds identified here were already known from oilseed rape. Exceptions are hexyl isobutyrate, *neo-allo*- and *allo*-ocimene, which were not previously known from oilseed rape [60,61,62,63,64]. Several of the floral scent compounds of oilseed rape, such as *p*-anisaldehyde, linalool and (*E*)-*β*-ocimene, are known as electrophysiologically and/or behaviorally active compounds in bee pollinators [10,20,24,64,65,66]. *p*-Anisaldehyde and linalool are also attractive to the cabbage seed weevil, *Ceutorhynchus assimilis* Payk., an important pest of oilseed rape [63,67].

The temperature affected neither the total amount of the scent released (Z= −0.93, N = 13, *p* = 0.353; optimum scenario: 0.29 ± 0.11 ng/flower/hour; warmer scenario: 0.57 ± 0.28 ng/flower/hour) nor the scent composition (Pseudo-F_1,12_ = 0.62, *p* = 0.729) between plants grown in the different temperature scenarios (Figure 1C,D). Independent of temperature, the scent of oilseed rape was dominated by *p*-anisaldehyde and linalool (Table 1, Figure 1D). Thus, the floral olfactory cues are likely not affected in this crop species, even if the temperature on earth increases by up to 5 °C until 2100. This species, however, seems to be more sensitive to other stressors, such as drought and ozone, which both affect floral scent signaling and have negative effects on pollinator attraction [68,69].

Overall, our results showed that buckwheat and oilseed rape are differently susceptible to heat stress and that increased temperatures resulted in smaller amounts and a different composition of scent only in buckwheat. In oilseed rape, the flower scent was independent of the temperature regime used for plant cultivation. The finding that different plants respond differently to increased temperatures is consistent with the data in the literature. Some studies have recorded decreased emissions and changes in the composition of floral scents under heat stress [26] (*Lilium auratum* Lindl.), [27] (*Globularia alypum* (L.), *Quercus ilex* L., *Dorycnium pentaphyllum* Scop., *Spartium junceum* L.) and [29] (*Jasminum auriculatum* Vahl), with one study even documenting that a plant species (strawberry, *Fragaria x ananassa* Duch) no longer produced detectable flower scents when exposed to heat stress [30], whereas other studies did not detect an effect of heat stress on floral scent emissions, [25] (*Trifolium repens* L.) and [70] (*Petunia axillaris* Lam.). Thus, some plants are more heat tolerant and resilient than others when it comes to signaling by floral scents.

However, even if plants are heat tolerant when it comes to olfactory communication with pollinators, high temperatures might affect traits relevant for plant reproduction other than floral scents [28], and plants are often exposed to other stressors in parallel, such as drought. For example, drought-stressed plants of buckwheat changed their floral scent compositions and were less attractive to pollinators, including *A. mellifera* and *Bombus* spp., than non-stressed plants [71]. The change in scent emission likely contributed to the reduced attraction of pollinators. Given that increased temperatures and drought also affect traits other than floral scents [28], future studies should also consider, in addition to signaling to pollinators, other traits (e.g., pollen tube growth, flower size, number of flowers) that might be influenced by increased temperatures and other environmental stressors to obtain a more holistic view of the effects of multiple stressors that simultaneously act on plant reproduction. Such studies might also consider how pollinators are affected by different stressors [72,73].

## 4. Conclusions

This study highlights that increased temperatures predicted by a global warming scenario differentially affect the floral scent emissions of crop plants since only one of the two studied crops was affected in the amount and composition of its scents. These changes affect the olfactory perception of the flowers by bee pollinators. Further studies are now needed to test whether these temperature-induced changes result in a different attractiveness of the flowers to their visitors, with potential effects on crop yields.

## Figures and Tables

**Figure 1 insects-14-00242-f001:**
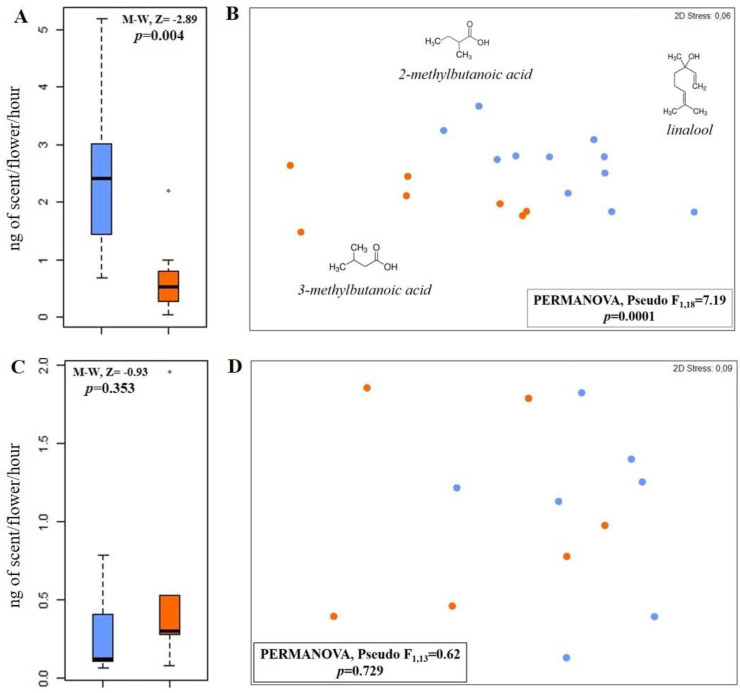
Total absolute amount of scent (ng of scent per hour and per flower) emitted in optimal (blue boxes) and warmer (orange boxes) scenarios by (**A**) buckwheat and (**C**) oilseed rape. Non-metric multidimensional scaling (NMDS) used to display semi-quantitative differences in scent profiles among scent samples collected in optimal (blue dots) and warmer (orange dots) scenarios from (**B**) buckwheat and (**D**) oilseed rape. Compounds indicated in (**B)** were most responsible for differences in scent composition between the two temperature scenarios, according to SIMPER analyses.

**Figure 2 insects-14-00242-f002:**
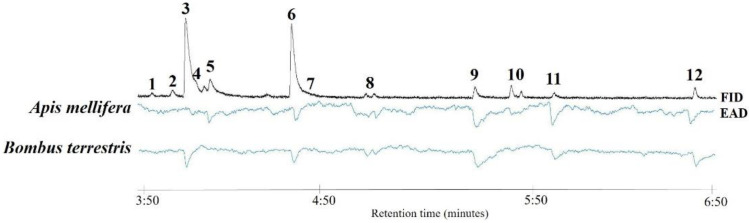
Examples of antennal responses (EAD) of *Apis mellifera* and *Bombus terrestris* to a mixture resembling the flower scent (FID) of buckwheat (*Fagopyrum esculentum*) in the optimum scenario. 1: sabinene; 2: *p*-benzoquinone; 3: 2-methylbutanoic acid; 4: butanoic acid; 5: (*E*)-*β*-ocimene; 6: 3-methylbutanoic acid; 7: linalool oxide furanoid; 8: linalool ((*R*)- and (*S*)-isomers); 9: *p*-cresol; 10: terpinene-4-ol; 11: *p*-anisaldehyde; 12: indole. Compounds not numbered are contaminants.

**Table 1 insects-14-00242-t001:** Total absolute amount of scent (ng of scent per hour and per flower; mean ± standard error) and relative amount (%) (± standard error) of the different compounds emitted at optimum and warmer scenarios of the studied crop species. Compounds are listed according to chemical class. KRI: Kovats Retention Index. The five highest relative amounts per species and treatment are in bold as are the names of buckwheat scents that elicited antennal responses in bee pollinators (see below). *: identification based on mass spectrum and retention index of authentic standard.

		Buckwheat		Oilseed Rape	
Total Absolute Amount of Scent		Optimum	Warmer	Optimum	Warmer
		Mean ± SE	Mean ± SE	Mean ± SE	Mean ± SE
		2.38 ± 0.39	0.71 ± 0.28	0.29 ± 0.11	0.57 ± 0.28
Compounds	KRI				
*Aliphatics*					
2-methylpropanoic acid *	742	0.8 (±0.2)	1.1 (±1.1)		
**butanoic acid** *	769	5.1 (±0.9)	2.6 (±1.3)		
pentanoic acid *	872	2.2 (±0.6)			
hexyl isobutyrate *	1146				0.1 (±0.1)
*Aromatics*					
***p*-benzoquinone** *	920	**6.8 (±1.1)**	**3.4 (±2.1)**		
***p*-cresol** *	1073	**7.8 (±3.7)**	3.2 (±0.2)		
2-methoxyphenol *	1095	0.1 (±0.1)			
1,4-dimethoxybenzene *	1168			**5.4 (±5.4)**	**6.1 (±0.5)**
***p*-anisaldehyde** *	1265	4.3 (±1.3)	1.3 (±1.3)	**40.8 (±13.0)**	**26.2 (±9.1)**
*p*-hydroquinone *	1267	0.2 (±0.1)			
*C5-branched chain compounds*					
**3-methylbutanoic acid** *	833	**28.0 (±2.8)**	**45.4 (±2.5)**		
**2-methylbutanoic acid** *	850	**18.0 (±1.9)**	**27.3 (±4.9)**		
*Monoterpenes*					
sabinene *	980	1.4 (±0.8)	**10.6 (±7.3)**		
*δ*-3-carene *	1016			**14.4 (±8.7)**	**11.2 (±4.8)**
*β*-phellandrene *	1037	0.1 (±0.1)			
(*Z*)-*β*-ocimene *	1038				3.4 (±3.4)
**(*E*)-*β*-ocimene** *	1050	3.5 (±1.3)	**3.9 (±3.9)**	**22.3 (±7.1)**	**13.0 (±8.4)**
**(*Z*)-linalool oxide furanoid** *	1078	0.6 (±0.2)	0.5 (±0.5)		
(*E*)-linalool oxide furanoid *	1094	0.1 (±0.1)			
**linalool** *	1102	**10.3 (±2.8)**		**17.1 (±7.9)**	**37.4 (±18.1)**
*allo*-ocimene *	1130				1.6 (±1.6)
1,3,8-*p*-menthatriene	1136	0.3 (±0.2)			
*neo*-*allo*-ocimene *	1143				1.0 (±1.0)
phenylacetonitrile *	1145	1.1 (±0.4)			
**terpinene-4-ol** *	1187	1.5 (±0.7)			
*Nitrogen-containing compounds*					
3-pyridinecarboxaldehyde *	1002	0.1 (±0.1)			
2-methylpyrazine *	1078	0.4 (±0.2)			
**Indole** *	1305	2.9 (±0.8)			
*Irregular terpenes*					
(*E*)-4,8-dimethyl-1,3,7-nonatriene	1119	4.4 (±1.1)	0.4 (±0.4)		

## Data Availability

The authors declare that the data supporting the findings of this study are available within the paper.

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
