# Peer review of "Floral Scents in Bee-Pollinated Buckwheat and Oilseed Rape under a Global Warming Scenario"

_insects, 2023, doi:10.3390/insects14030242_

Round 1

Reviewer 1 Report

This research is a significant contribution in the field of climate change, impacts of global warming on plants and their interacting pollinators. Manuscript is written well. Results and discussions are comprehensively formatted. Some suggestions have been given for the improvement in the attached file.

Author Response

TITLE

It may be revised by mentioning the names of crops in the title as this title looks too much generalized

Done, the title now reads: Floral scents in bee pollinated buckwheat and oilseed rape under a global warming scenario.

ABSTRACT:

  1. Line 23 must be started with the introductory lines relevant to title either mentioning about the importance of crops, then their pollinators. After this authors must come towards the emission of floral scents.

We rephrased line 23 and the next sentences as suggested.

  1. Line 39 must end with the future prospects of these studies. Give a single line statement of gaps to be filled by future scientists.

We added a sentence with suggestions for further studies.

INTRODUCTION

This part seems too short. It must be increased after improving the title and then keeping in mind the results.

It was improved by adding some sentences and references.

  1. Line 45 if available kindly update citations

We added an updated reference.

  1. Line 55-56 seem confusing kindly rewrite these line.

We rewrote these lines.

  1. After line 59 you must give a mini review about the production of different compounds, scents produced in different crops at different temperatures and response of various pollinators in previous studies.

We added some sentences about the production of floral scents in different crops, however, the crop floral scents produced at different temperatures and the responses of the pollinators are still unknown. Therefore, our study fills part of this gap.

  1. Line 72 -78 are the parts of summary and abstract. They must be written again.

We rephrased these lines.

MATERIAL AND METHODS

  1. Line 81-85 must be the part of introduction in the beginning.

We shifted these lines to Introduction section.

  1. Line 90-96 you must cite the methodology followed.

Regarding the methodology here, the plant growth-chambers were adjusted by technicians of the University, therefore, we described it in detail and did not use a citation. One reference was added concerning how the plant cultivation was performed. 

  1. Line 122-143, 160-169 kindly cite the methodology

Done.

Results and Discussion

  1. Line 234-236 need correction and written again please.

Done.

  1. Line 249-250 must be added after this part in conclusion as suggestions for future studies

Done.

  1. Line 261-262 sentence (and most of the here identified compounds were already known to oilseed rape.) need correction

Done.

  1. After line 309 you must give a conclusion

We added a conclusion section.

Reviewer 2 Report

The authors conducted an interesting study about increased temperature consequences on oilseed rape and buckwheat flower sents. Then they question the capability of Bombus terrestris and Apis mellifera to detect these changes in flower scents. They clearly explain their protocols for flower scents analyses, discuss the results, and provide perspectives to conduct further research based on the presented results. 

For me, the weakest point is the introduction, which needs to better summarise state of the art about entomophilous pollination and insect diversity and attractiveness. 

Keywords: "floral scent" is already in the title, so it may not be useful in the keyword section.

L.45 "Pollination by bees" is restrictive, isn't it? Indeed, it is pollination by insects that is an essential ecosystem service…

L.78 are Apis mellifère and Bonus terrestris really the main pollinators? Or only the main managed pollinators? I would choose the second option…

L.85 other studies, like Ouvrard et al. 2017 https://doi.org/10.2135/cropsci2016.09.0735, report that insects are negligible pollen vectors for oilseed rape seed production.

L. 86, Klein and all . 2007 did not report Bombus terrestrial nor Apis mellifère as the main pollinators of oilseed rape or buckwheat.

Fig. 2 what is the line and numbers below the figure?

L.320, you refer to supplemental files, but you did not provide them with the manuscript for review.

L.323, it is nice to thank the team of Botanical Garden for help with strawberry plants, but what is the link with the current paper, as you did not use strawberry plants?

Author Response

For me, the weakest point is the introduction, which needs to better summarise state of the art about entomophilous pollination and insect diversity and attractiveness.

The state of the art of these topics was added to the Introduction. 

Keywords: "floral scent" is already in the title, so it may not be useful in the keyword section.

It was removed.

L.45 "Pollination by bees" is restrictive, isn't it? Indeed, it is pollination by insects that is an essential ecosystem service…

It is true and this sentence was improved.

L.78 are Apis mellifère and Bonus terrestris really the main pollinators? Or only the main managed pollinators? I would choose the second option…

Yes, both bee species are the main pollinators of the target crops (see Björkman 1995, Klein et al. 2007, Woodcock et al. 2013). We added some references to reinforce the role of Bombus terrestris as pollinator of buckwheat. However, both bee species are also managed, therefore we added “managed” to the sentence.

L.85 other studies, like Ouvrard et al. 2017 https://doi.org/10.2135/cropsci2016.09.0735, report that insects are negligible pollen vectors for oilseed rape seed production.

Thank you for this reference, we added it.

L. 86, Klein and all . 2007 did not report Bombus terrestrial nor Apis mellifère as the main pollinators of oilseed rape or buckwheat.

In the Supplementary materials from Klein et al. (2007), Apis mellifera is pointed out as the main pollinator of oilseed rape and buckwheat, and Bombus species only of oilseed rape, therefore, we added some references that reported Bombus species as buckwheat pollinator.

Fig. 2 what is the line and numbers below the figure?

It represents the retention time. This information was added to the figure.

L.320, you refer to supplemental files, but you did not provide them with the manuscript for review.

It was removed.

L.323, it is nice to thank the team of Botanical Garden for help with strawberry plants, but what is the link with the current paper, as you did not use strawberry plants?

It was corrected.